

# Pots vs trammel nets: a catch comparison study in a Mediterranean small-scale fishery

Andrea Petetta[1,2,*], Claudio Vasapollo[2,*], Massimo Virgili[2], Giada Bargione[1,2] and Alessandro Lucchetti[2]

[1] Department of Biological, Geological and Environmental Sciences, University of Bologna, Bologna, Italy
[2] Institute for Biological Resources and Marine Biotechnologies (IRBIM), Italian National Research Council (CNR), Ancona, Italy
* These authors contributed equally to this work.

Corresponding authors
Andrea Petetta,
andrea.petetta@irbim.cnr.it
Alessandro Lucchetti,
alessandro.lucchetti@cnr.it

## ABSTRACT

Passive bottom-set nets are the most widely used fishing gears in Mediterranean small-scale fisheries (SSFs). Trammel nets, in particular, have key advantages such as their ease of use and handling and high capture efficiency for numerous commercial species. However, they entail high discard rates (5–44% of the total catch) connected to high mortality, thus exerting an adverse impact on benthic communities, besides catching individuals of commercial species under the minimum conservation reference size (MCRS) and specimens of protected species. Fish pots are seen as alternative and a more sustainable gear type that allow reducing discards in SSFs. In this study, a collapsible pot was tested at three coastal sites in the north-western Adriatic Sea (GFCM GSA 17) to compare its catch efficiency with that of the local traditional trammel nets. Data analysis demonstrated a similar catch efficiency for the commercial species, both among sites and as a whole. Moreover, the trammel net caught a larger amount of discards, both in terms of species number and of CPUE_W. The catch comparison study involved the two most abundant landed species, common cuttlefish *Sepia officinalis* and annular sea bream *Diplodus annularis*. The pots were more effective for *S. officinalis*, whereas the trammel net was more effective for the shorter length classes for *D. annularis*, which were mostly under the MCRS (12 cm). The innovative pots could provide a valuable alternative to the trammel nets traditionally used in the Adriatic Sea, at least in certain areas and periods. Their main advantages include that they do not require a different rigging and they can be used without bait, while their foldable design allows large numbers to be easily loaded on board SSF vessels. The results of this pilot study indicate that pots can achieve the objectives of reducing discards and bycatch in SSFs without penalizing the catch of commercial species.

## INTRODUCTION

Gillnets and trammel nets (or set/passive nets) are the most widely used fishing gears in Mediterranean small-scale fisheries (SSFs) (*Lucchetti et al., 2015*), which account for more than 70,000 vessels (*FAO, 2018*) and approximately 150,000 jobs. Set nets consist of netting panels hanging in the water column, where they are held perpendicular to the bottom by floaters and sinkers. Bottom-set fixed nets passively exploit the movements of target species (*Gabriel et al., 2005*). The fish swimming into them are caught by being gilled, tangled or wedged in gillnets, which are constituted of a single netting panel. On the contrary, the typical catching method of trammel nets is trapping the fish in a pocket of netting thanks to three panels: an inner panel with small mesh size and two outer panels with larger mesh size (*Fabi et al., 2002*). The success of the passive nets is due to their ease of use and handling (especially on small boats) (*Dinçer & Bahar, 2008*), high selectivity (especially gillnets) (*Holt, 1963*; *Fabi et al., 2002*) and their high capture efficiency for numerous commercial species (*Amengual-Ramis et al., 2016*). Their technical parameters, for example, mesh size, netting twine, hanging ratio and net drop, vary widely in relation to the characteristics of target species and fishing areas (e.g., depth, seafloor), as do their selection properties (*Stergiou et al., 2006*; *Lucchetti et al., 2017*). Although passive nets are considered as selective gears, they nonetheless produce a large amount of discards (*Goncalves et al., 2007*; *Tzanatos et al., 2007*) that range from 5% to about 40% of the total biomass caught (*Tsagarakis, Palialexis & Vassilopoulou, 2014*). SSFs discards consist of species with low commercial value, individuals that are found in poor condition and specimens under the minimum conservation reference size (MCRS; *Regulation (EU), 2019*). The proportion of undersized individuals in the catch is variable and for some commercial species it can be quite high (20% for *Sparus aurata*, 28.2–74.8% for *Diplodus* spp., 93.8% for *Pagellus acarne* in the eastern Mediterranean, *Tzanatos et al. (2008)*; large numbers of *Diplodus bellottii*, *Argyrosomus regius* in Cadiz and *Diplodus* spp., *Pagrus pagrus* in the Cyclades, *Goncalves et al. (2007)*). Notably, species that are caught in excessively small amounts for the fishers' target market may also become discards (*Goncalves et al., 2007*). Moreover, set nets can be also responsible for the incidental catch of protected species such as sea turtles (*Lucchetti & Sala, 2010*; *Casale, 2011*; *Lucchetti, Vasapollo & Virgili, 2017a*, *2017b*) and elasmobranches with no economic value (*Morey et al., 2006*; *Saidi, Enajjar & Bradai, 2016*; *Bradai, Saidi & Enajjar, 2018*).

The reduction of discards and bycatch has become a priority for fisheries worldwide, by means of measures to improve selectivity and to preserve the environment (*FAO, 2011*). The Common Fishery Policy, through the article 15 of *Regulation (EU) (2013)*, calls for the development of more selective technical solutions, to avoid the catch of unwanted species and sizes. Several solutions are being tested in the Mediterranean. They include: (i) gear modifications to improve size and species selection (*Lucchetti et al., 2015*, *2017*); (ii) time/area fishing closures to minimize bycatch (*Lucchetti, Vasapollo & Virgili, 2017a*); (iii) mitigation devices to avoid catching some protected species (e.g., UV lights for sea turtles; *Virgili, Vasapollo & Lucchetti, 2017*); and (iv) alternative and more sustainable fishing gears (*Amengual-Ramis et al., 2016*).

As regards the latter point, experimental pots developed in the past few years in certain areas have ensured catch efficiencies comparable to those of traditional set nets (*Furevik & Hågensen, 1997*; *Iskandar et al., 2006*; *Furevik et al., 2008*; *Königson et al., 2015*; *Amengual-Ramis et al., 2016*). Pots are passive gears to which fish, crustaceans and mollusks are attracted by bait or pasture, whereas cephalopods are caught because use them as a refuge or a site to spawn. Pots have several appealing features—in particular a minimal habitat impact and low manufacturing cost—which have led them to be classified as LIFE (low-impact fuel-efficient) gears (*Suuronen et al., 2012*). Moreover, bycatch can be minimized by acting on bait, mesh size, materials, and position/design of the entrance and the escape gap(s) (*Furevik & Løkkeborg, 1994*; *Furevik & Hågensen, 1997*; *Boutson et al., 2009*).

In Mediterranean SSFs traditional pots are locally employed to target mollusks and crustaceans (*Grati et al., 2010*; *Amengual-Ramis et al., 2016*), ensuring high catch efficiency and low discard rates (0.8–6.6%; *Fabi et al., 2001*). They are usually deployed in specific seasons (e.g., the spawning period of *Sepia officinalis*, *Melli et al., 2014*), in circumscribed areas (e.g., north western Adriatic Sea for *Squilla mantis*, *Grati et al., 2018*; Gulf of Càdiz (Spain), Thracian Sea (Greece) and Gulf of Gabès (Tunisia) for *Octopus vulgaris*, *Ezzeddine-Najai, 1992*; *Tsangridis, Sánchez & Ioannidou, 2002*; *Sobrino et al., 2011*), or in replacement of other gears (e.g., for *Nephrops norvegicus* during trawl fishing closures in Croatian northern Adriatic waters, *Brčić et al., 2018*). A major disadvantage of traditional pots is their large volume, which entails that vessels can carry only a limited number of units per trip.

In the Mediterranean Sea, studies of alternative fishing gears such as innovative pots are still limited (*ICES, 2008*, *2009*; *Pol, He & Winger, 2010*) and mainly regard those targeted to cephalopods like *O. vulgaris* (*Sbrana et al., 2008*) and crustaceans such as *N. norvegicus*, *Plesionika* spp. (*Colloca, 2002*; *Sartor et al., 2006*) and *Palinurus elephas* (*Amengual-Ramis et al., 2016*).

Based on these considerations, a pilot study was devised to test a fully collapsible pot design and to compare it to a traditional set net in commercial fishing conditions. The main goals of the study were to evaluate the respective catch compositions and to assess the effectiveness of the pots in terms of their use and handling, discards and bycatch reduction.

## MATERIALS AND METHODS

### Study area

The pilot study was conducted in FAO Geographical Sub-Area 17 (north-western Adriatic Sea) and involved three coastal areas (Marina di Ravenna, Portonovo and Senigallia), where depth ranges from 5 to 19 m (Fig. 1). Specifications of bottom type and average depth of the three sites are listed in Table 1. In these areas, SSFs mostly employ gillnets and trammel nets to catch cuttlefish (*S. officinalis*), various fish species (e.g., *Solea solea*, *Lithognathus mormyrus*, *Diplodus* spp*.*, *S. aurata*, *Sciaena umbra*, *Umbrina cirrhosa*, *Dicentrarchus labrax*) and crustaceans (*S. mantis*, *Paeneus kerathurus*) (*Fabi & Grati, 2005*). Moreover, artisanal pots are deployed in spring to specifically target cuttlefish.
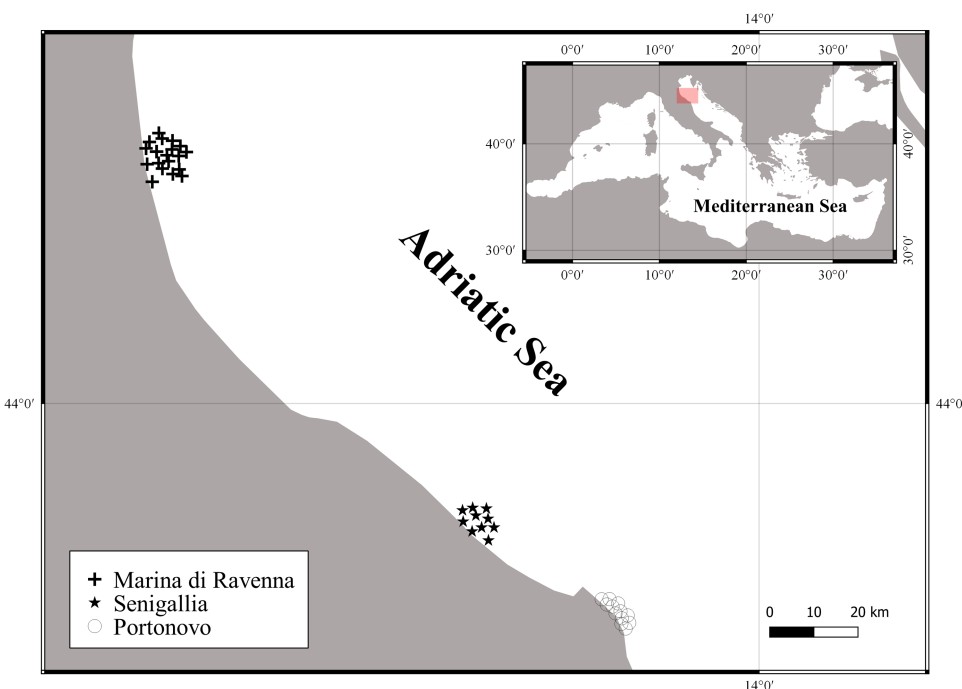

**Figure 1 Map of the study area where the trials were performed in 2016 and 2017 (April–August).**

**Table 1 Summary of the fishing trials carried out at the three sites (Marina di Ravenna, Senigallia, Portonovo).**

|  | Marina di Ravenna | Senigallia | Portonovo |
|---|---|---|---|
| Bottom type | Sandy-mud with scattered rocky outcrops | Sandy-mud | Rocky |
| AVG depth (m) ± SD | 9.9 ± 2.5 | 10.7 ± 0.43 | 6.0 ± 0.57 |
| Vessel characteristics | LOA 12.4 m; 10 GT; 350 kW | LOA 12.4 m; 6 GT; 130 kW | LOA 6.6 m; 1 GT; 100 kW |
| Study period | April-July 2017 | April-June 2016 | May-August 2016 |
| No. of trials | 20 | 10 | 12 |
| GTR length (m) | 500 | 300 | 500 |
| No. of LPs | 20 | 9-10 | 0 |
| No. of SPs | 20 | 19-20 | 20 |
| AVG GTR soak time (h) ± SD | 19.9 ± 3.3 | 17.4 ± 2.0 | 17.2 ± 1.8 |
| AVG LP soak time (h) ± SD | 91.4 ± 28.7 | 73.9 ± 9.1 | – |
| AVG SP soak time (h) ± SD | 87.3 ± 25.6 | 76.6 ± 10.1 | 90.1 ± 21.3 |

**Note:**
AVG, average; SD, standard deviation; LOA, length all out; GT, gross tonnage; GTR, trammel net; LP, large pots; SP, small pots.

## Fishing gears and experimental setup

The trials were conducted on board local professional fishing vessels (Table 1).
The characteristics of the traditional trammel nets (GTRs) employed and the fishing

**Figure 2 Scheme of the trammel net used in the study.** Inner panel: middle; outer panel: top and bottom. PA, polyamide; PP, propylene; ø: diameter; E: hanging ratio.

grounds were selected by the fishers, and similarly the fishing operations (e.g., fishing time) and the sorting of the catch were carried out following the fishers' procedures, without interferences from the scientists on board. Experiments to compare the GTRs and the foldable pots were carried out from April to August, in 2016 and 2017, and involved a total number of 42 fishing trials: 20 at Marina di Ravenna, 12 at Portonovo and 10 at Senigallia. To minimize differences due to patchy species distribution, GTRs and pots were deployed close to each other (a few tens of meters).

The technical features of the GTRs are reported in Fig. 2. The three netting panels were made of transparent polyamide multifilament: 210/4 mm multifilament and 36 mm mesh bar for the inner panel and 210/3 mm multifilament and 200 mm mesh bar for the outer panels. The net had a nominal height of 2.5 m (35 meshes), although its effective vertical opening in the water was around 1.5 m. The float line and lead line were in propylene (diameter, 8 and 10 mm, respectively); the float line was reinforced with external oval-shaped floats (diameter, 8 cm); the lead line weighed 120 g/m. The total length of the set nets used in each site was: 500 m at Marina di Ravenna and Portonovo, 300 m at Senigallia (Table 1). The GTRs were set early in the morning and hauled in the afternoon.

The pots (manufactured by Trapula Ltd., Croatia; Fig. 3) have a stainless-steel bar frame with a pentagonal shape and a single oval entrance. Two steel structures on the top and bottom allow folding them. A propylene rope 5 mm in diameter was externally reinforced with a nylon net (32 mm square-mesh bar). Flexible steel bars 2 mm in diameter allow manual adjustment of the opening. To establish whether catch efficiency was related to the volume of the chamber (*Furevik & Løkkeborg, 1994*), two different pot sizes were tested: a smaller pot (SP) measuring 40 × 100 cm (height × width) and a larger pot (LP) measuring 60 × 140 cm. The pots were attached to a main propylene line (namely a gang) 8 mm in diameter anchored to the seabed by 2 m plastic branch lines 5 mm in

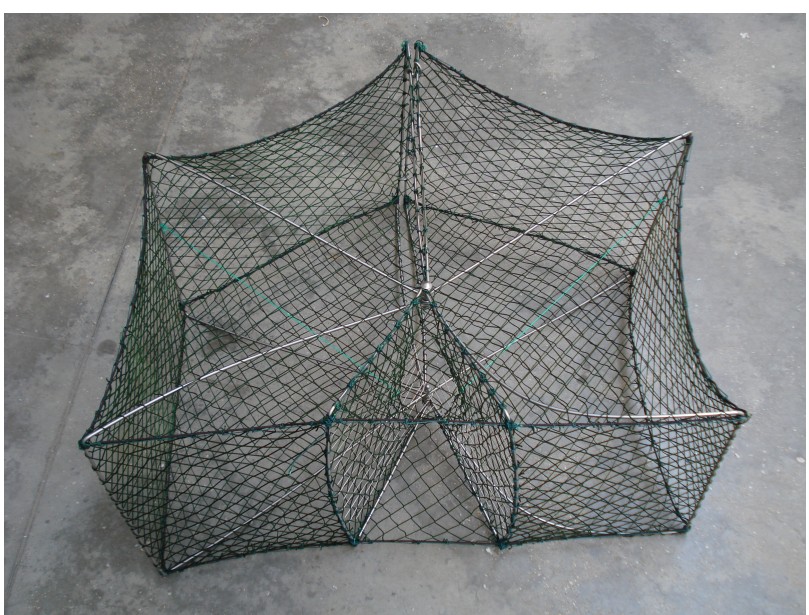

**Figure 3 Picture of the "Trapula" pots tested in the study.** See text for dimensions.

diameter, placed at 15 m intervals. The pots were set 15 m apart according to the traditional rigging used by the local fishers (*Fabi et al., 2001*). The number of pots of each type ranged from 9 to 20 per fishing trial. Soaking time (i.e., the period of time the pots were left on the bottom) depended on weather conditions and fishers' tactics (Table 1). They were commonly retrieved after 2–3 days, or more, in case of adverse weather conditions, to attract a wider range of commercial species other than cuttlefish, for which traditional pots are usually left 24 h (*Fabi, 2001*). The pots were not baited, but several black plastic ribbons were attached to the frame, to attract cuttlefish.

The GTR and pot hauls were paired: each haul consisted in setting and retrieving the GTR, the gang with SPs and the gang with LPs.

## Data collection

For each haul, the crew sorted onboard the catch, that was kept separate by gear (GTR, LP, SP). The total catch for each gear was thus divided into a landed catch (species with commercial value, not necessarily target species) and discards, that is, species discarded for different reasons (invertebrates and fish species with no commercial value, commercial individuals under the MCRS or in poor conditions). All individuals were identified to the lowest taxonomic level possible, counted, weighed to the lowest 0.1 g and measured to the nearest 0.5 cm for total length (TL; fish) or mantle length (ML; cephalopods) and to the nearest 0.5 mm for carapace length (CL; crustaceans).

## Data analysis

The catch per unit effort (CPUE) was calculated for GTRs and pots. The GTR catch was standardized for the number of individuals ($CPUE_I$) and total catch weight ($CPUE_W$;

in kg) captured by 1,000 m of net in 12 h, considering that fishers commonly set 3,000 to 6,000 m of net and haul it up after about half a day. The pot catch was standardized for the $CPUE_I$ and $CPUE_W$ (in kg) captured by 66.6 pots (i.e., the number of pots corresponded to 1,000 m of set net considering 15 m distance between two subsequent pots, according to *Fabi et al. (2001)*) in 24 h. This duration corresponds to the commercial fishing time of the traditional pots used in the area to target cuttlefish (*Fabi, 2001*). The Kruskal–Wallis $H$ test ($\chi^2$) was applied to seek differences between the $CPUE_W$ of GTRs and pots. A non parametric test was adopted, because the data distributions were not normal and extremely skewed, with wide tails. If differences did emerge, a pairwise Wilcoxson's signed rank test based on Bonferroni correction for multiple comparisons was applied to establish the levels showing significantly different median values.

Differences in the size of the individuals caught by the GTRs, LPs and SPs were explored by analyzing the length frequency distribution (LFD) of the landed species. The catch efficiency of each pot type vs GTR was compared using generalized linear mixed models (GLMMs; *Holst & Revill, 2009*). The probability for an individual to be retained in a pot follows from:

$$Pr\{Pot/(Pot + GTR)\} = 1 \bigg/ \left(1 + e^{-(\beta_0 + \beta_1 \times length + \beta_2 \times length^2 + \beta_3 \times length^3)}\right)$$

A binomial error distribution was used to calculate the probability of the number of fish caught in a pot ($CPUE_I$) given that they were caught by both gears based on 1-cm size classes. A probability value of 0.5 corresponds to equal catches in both gears. According to *Holst & Revill (2009)*, a third order polynomial would be adequate for most cases, although in some instances a first or second order would be enough. The best binomial model was chosen based on the lowest Akaike's Information Criterion (AIC) value. A random term was added to the models. Since the GTR and pot hauls were paired, the catches for each site and for each gear were pooled and the term "site" was used as a random intercept.

The most abundantly caught species, *S. officinalis* and *Diplodus annularis*, were selected for the catch comparison analysis. Since only SPs were set at Portonovo, the catch comparison of LPs included only the Senigallia and Marina di Ravenna. The models are illustrated graphically with a 95% confidence interval (CI) calculated with a bootstrap method using 999 simulations. The free software R (*R Core Team, 2018*) and the R packages nlme (*Pinheiro et al., 2018*) and lme4 (*Bates et al., 2014*) were used for the analyses.

## RESULTS

Overall, the three gears caught 53 species, 38 of which belonged to the landed fraction (GTRs = 30, LPs = 15 and SPs = 22) and 28 to the discard species (GTRs = 25, LPs = 5 and SPs = 5), thus confirming that the pots were more species-selective than GTRs (Tables S1 and S2).

As regards the landed species, cuttlefish (*S. officinalis*) was the most abundant in terms of biomass at all 3 sites for all three gears. Two other abundant species caught by all

**Table 2 Mean biomass values (CPUE$_W$) with standard errors and confidence intervals, in brackets, of the landed catch and of discards for the three gears.**

| | Site | GTR CPUE$_W$ | LP CPUE$_W$ | SP CPUE$_W$ |
|---|---|---|---|---|
| Landed catch | M. di Ravenna | 5.41 ± 0.76 (3.92–6.89) | 7.27 ± 1.77 (3.81–10.73) | 3.72 ± 0.64 (2.48–4.97) |
| | Senigallia | 4.34 ± 0.14 (2.16–6.52) | 2.41 ± 0.35 (0.23–4.59) | 3.08 ± 0.17 (2.20–3.97) |
| | Portonovo | 2.90 ± 0.70 (1.53–4.27) | – | 2.48 ± 0.59 (1.33–3.62) |
| Discards | M. di Ravenna | 0.77 ± 0.20 (0.38–1.17) | 0.14 ± 0.04 (0.07–0.21) | 0.07 ± 0.05 (0.02–0.17) |
| | Senigallia | 0.95 ± 0.57 (0.17–2.07) | – | 0.04 ± 0.0 (NA) |
| | Portonovo | 1.52 ± 0.79 (0.02–3.06) | – | 0.06 ± 0.0 (NA) |
| | **Gear** | **CPUE$_W$** | | |
| Landed catch | GTR | 4.35 ± 0.51 (3.35–5.34) | | |
| | LP | 6.01 ± 1.39 (3.28–8.74) | | |
| | SP | 3.21 ± 0.36 (2.49–3.93) | | |
| | GTR | 0.95 ± 0.23 (0.51–1.40) | | |
| Discards | LP | 0.14 ± 0.04 (0.06–0.21) | | |
| | SP | 0.06 ± 0.03 (0.01–0.12) | | |

**Note:**
GTR (trammel net), LP (large pot) and SP (small pot), at each site (top) and as a whole (bottom). NA (not available).

three gears were annular seabream (*D. annularis*), caught at Marina di Ravenna and Senigallia but not at Portonovo, and striped seabream (*L. mormyrus*), caught at all three sites with greater abundance at Senigallia and Portonovo. Additional landed species caught by GTRs were *S. solea*, *S. mantis* and *S. umbra* at Marina di Ravenna; *Liza aurata*, *Sarda sarda* and *Scophthalmus rhombus* at Senigallia; and *Mugil cephalus* at Portonovo. Other landed species caught by the pots were *S. umbra* (LP, SP) and *Conger conger* (LP) at Marina di Ravenna and *Dentex dentex* (SP) at Portonovo (Table S1).

The mean biomass values (calculated as CPUE$_W$) and 95% CIs of the landed catch are reported in Table 2. The CI values of the three gears did overlap, indicating the lack of significant differences among them, both at each site and as a whole. Standardization of the landed catch weight failed to highlight significant differences in the medians among GTRs, LPs and SPs, either within the three sites or as a whole ($\chi^2$ = 2.59, df = 2, $p$ = 0.274; Fig. 4).

The discards of LPs and SPs were lower than those of the GTRs both in terms of species number and of CPUE$_W$; in fact, they were close to zero both at Senigallia and at Portonovo (Table S2). The GTRs captured large amounts of *Alosa fallax* and *Pteroplatytrygon violacea* at Marina di Ravenna; *A. fallax* and *Liocarcinus vernalis* at Senigallia and *Eriphia verrucosa*, *Hexaplex trunculus* and *Maja crispata* at Portonovo (Table S2).

The CIs of the mean biomass values (CPUE$_W$) of GTR discards did not overlap with those of LPs and SPs, whereas those of LPs and SPs did (Table 2). The discards showed significant differences in terms of standardized biomass ($\chi^2$ = 11.34, df = 2, $p$ = 0.004, Fig. 4) and were mostly caught by GTRs. Wilcoxson's pairwise test showed that the median

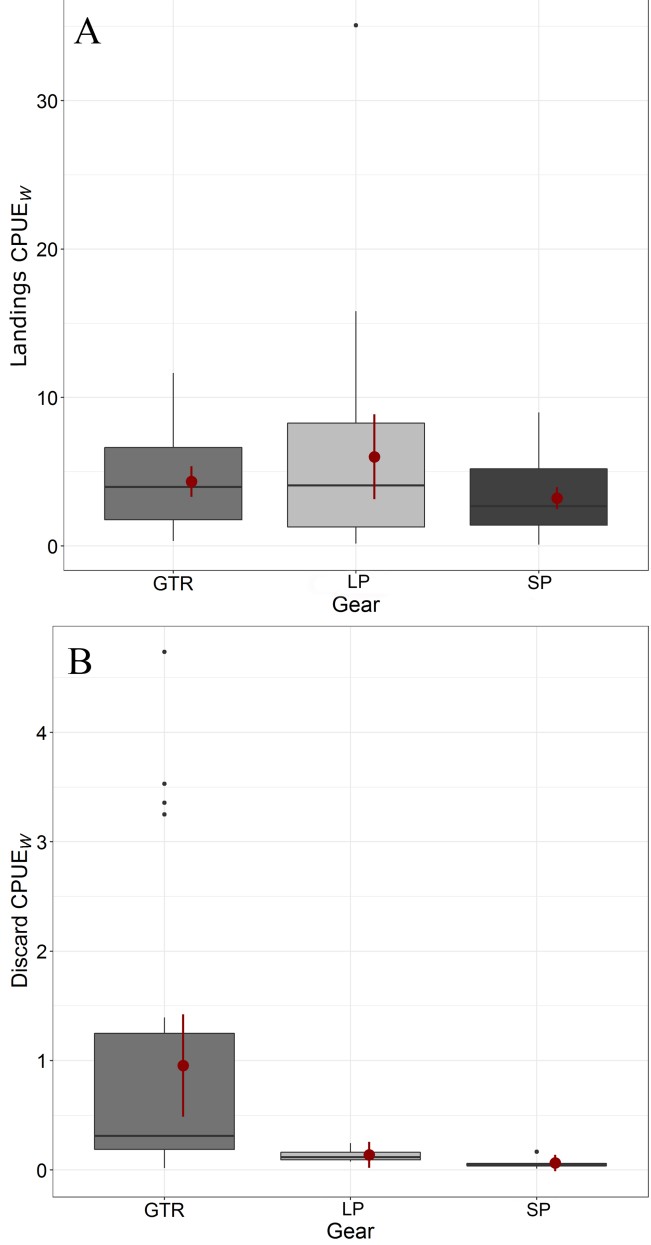

**Figure 4 Overall CPUE$_W$ of landings and discards of the three gears tested in the study.** GTR: trammel nets; LP: large pots; SP: small pots. Red dots: mean CPUE$_W$; red bars: confidence intervals (CIs). (A) Commercial; (B) Discard.                              

differences between gears were significant for GTRs vs LPs and for GTRs vs SPs ($p = 0.011$ and $p = 0.016$, respectively), whereas the medians of LPs and SPs were not significantly different.

The LFD of *S. officinalis* and *D. annularis* at each site and as a whole is reported in Fig. 5. The lines of LPs and SPs were mostly above those of GTRs, indicating a greater catch efficiency. The catch comparison curves (Fig. 6; Table 3) demonstrate that for *S. officinalis*, pots (both dimensions) were more efficient than GTRs. SPs were more efficient than

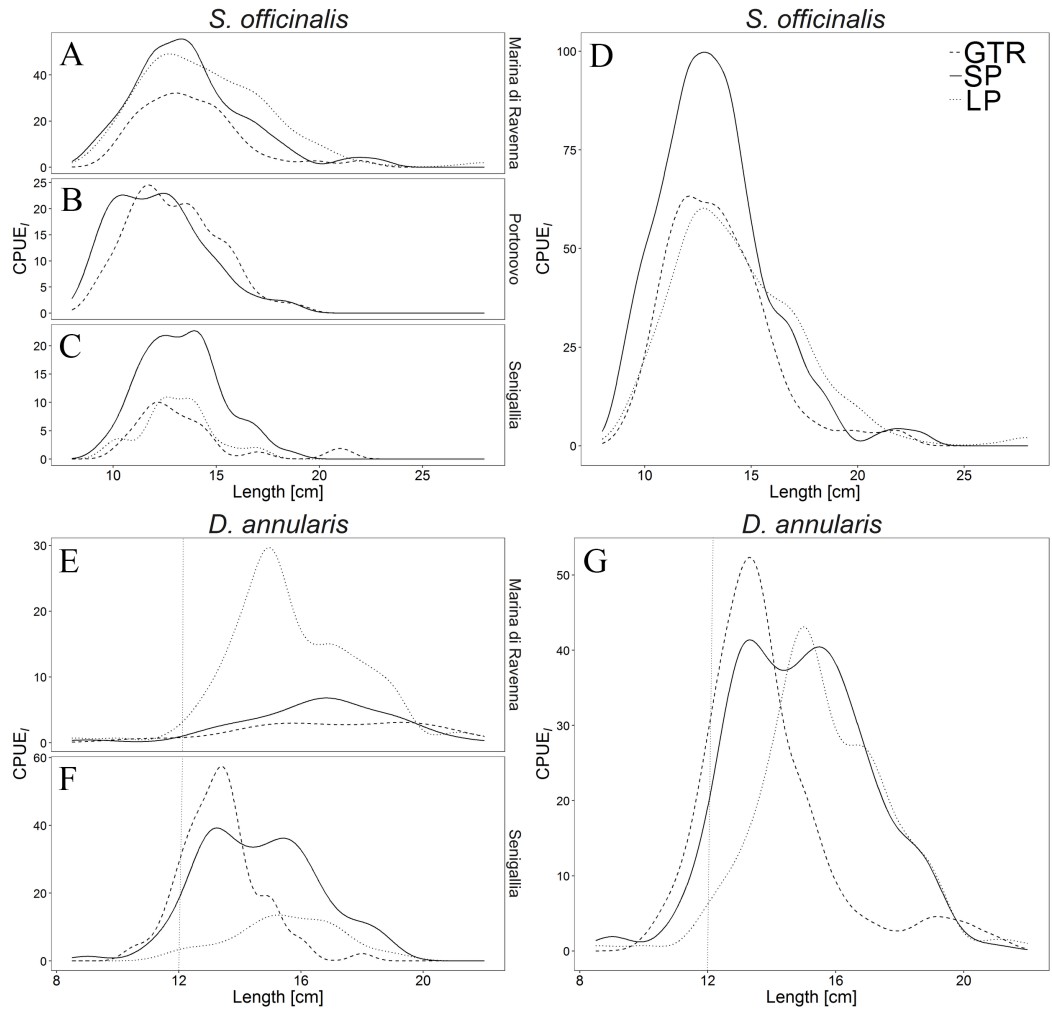

**Figure 5 Length frequency distributions (LFDs) of *Sepia officinalis* in each site (A, B and C) and as a whole (D) and LFDs of *Diplodus annularis* in each site (E and F) and as a whole (G).** (A and E) Marina di Ravenna; (B) Portonovo; (C and F) Senigallia. Dashed lines represent GTR (trammel nets); continuous lines represent SP (small pots); dotted lines represent LP (large pots); vertical dotted lines represent MCRS of 12 cm of *D. annularis*.

GTRs for most *S. officinalis* sizes, except for the larger ones (above the 25 cm size class), for which the efficiency of both gears were similar, as the lower CI exceeds the limit of 0.5 indicating equal proportion of individual catches between both gears. In contrast, LPs showed the same efficiency as GTRs for the smaller individuals (below the 11 cm size class) and were more efficient for the larger individuals. LPs and SPs showed overlapping CIs from the 10 cm size class, i.e. a similar catch efficiency. As regards *D. annularis*, the GTR reflected a greater efficiency than both LPs and SPs at the smaller sizes, usually under the MCRS of the species (12 cm). As a result, the percentage in number of undersized individuals of *D. annularis* caught by GTR was 15.1% (7.1% for Marina di Ravenna; 16.4% for Senigallia), while it was lower for both pots: 7.3% for SPs (3.3% for Marina di Ravenna; 8% for Senigallia) and 3.7% for LPs (3.3% for Marina di Ravenna; 4.5% for Senigallia).

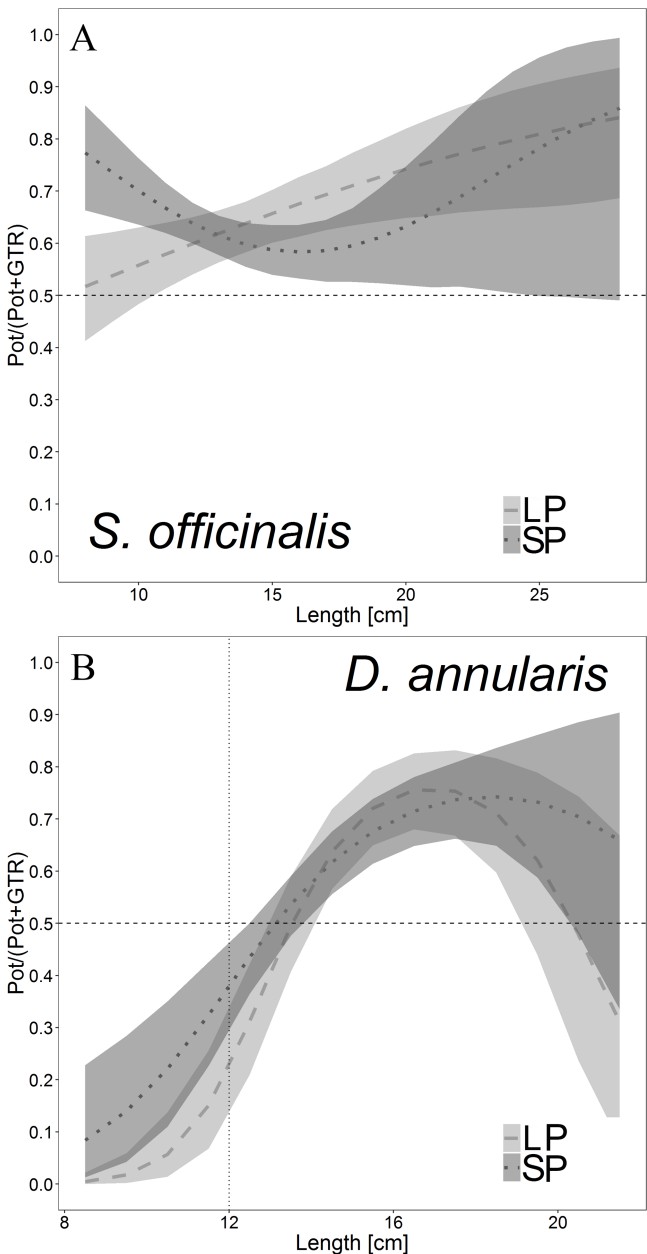

**Figure 6 Catch comparison curves for *Sepia officinalis* (A) and *Diplodus annularis* (B), representing the GLMM proportions of the total catches of the three gears.** GTR, trammel nets; LP, large pots; SP, small pots. Dashed and dotted lines represent the mean proportions of LP and SP, respectively; the vertical dotted line represents MCRS of 12 cm of *D. annularis*. Interpretation: a value of 0.5 indicates an even split between GTRs and Pots, whereas a value of 0.25 indicates that the Pots caught 25% of all the fish of that length class whereas 75% were caught by the GTRs. Shaded areas: 95% confidence intervals.

Therefore, LPs and SPs both showed a high selectivity for larger fish, although they also caught some undersized individuals. However, there were no differences among the three gears in the catch efficiency for the larger *D. annularis* individuals.

**Table 3 Estimates of the parameters of the GLMM calculated for catch comparison.**

| Species | Model | | Parameter | Estimate | SE | $p$ |
|---|---|---|---|---|---|---|
| *Sepia officinalis* | Linear | LP vs GTR | $\beta_0$ | −0.61 | 0.45 | 0.171 |
| | | | $\beta_1$ | 0.08 | 0.03 | 0.009 |
| | Quadratic | SP vs GTR | $\beta_0$ | 4.07 | 1.47 | 0.006 |
| | | | $\beta_1$ | −0.46 | 0.20 | 0.022 |
| | | | $\beta_2$ | 0.01 | 0.01 | 0.038 |
| *Diplodus annularis* | Quadratic | LP vs GTR | $\beta_0$ | −27 | 5.62 | <0.001 |
| | | | $\beta_1$ | 3.33 | 0.71 | <0.001 |
| | | | $\beta_2$ | −0.98 | 0.02 | <0.001 |
| | Quadratic | SP vs GTR | $\beta_0$ | −11.39 | 4.24 | 0.007 |
| | | | $\beta_1$ | 1.35 | 0.56 | 0.017 |
| | | | $\beta_2$ | −0.04 | 0.02 | 0.048 |

**Note:**
SE, standard error; GTR, trammel net; LP, large pots; SP, small pots.

## DISCUSSION

This study was aimed at testing the catching efficiency of an innovative pot design in the north-western Adriatic Sea; in particular, it was evaluated the pot's ability to provide an alternative to traditional trammel nets and to reduce discards in SSFs. The comparison implied a different standardization ($CPUE_W$ at 12 h for trammel nets and 24 h for pots) that represented a compromise to maintain the standardization unit as close as possible between the two gears, taking into account that they operate in different ways and with a different fishing time. The main finding, from the catch comparison analysis of the two most abundant species caught (cuttlefish and annular sea bream), was that the catch rates of the two types of pot tested, which differed only in dimensions, were comparable to those of the trammel nets. These data are in line with the high efficiency of the experimental pots reported in the Barents Sea (*Furevik et al., 2008*) and the Baltic Sea (*Königson et al., 2015*), where pots showed similar, if not higher, catch rates than those of passive nets, at least in a period of the year. Interestingly, the innovative pots caught a larger number of commercial species than the traditional ones used in the area by artisanal fishers, which are not collapsible, with a different shape and entrance, and mainly target cuttlefish (*Fabi et al., 2001*).

Regarding the annular sea bream, the poor selectivity of the trammel nets found for this species has been previously reported for *Diplodus* spp. by *Tzanatos et al. (2008)*, who estimated a percentage (in weight) of the undersized individuals caught of 28.2% for *D. annularis* and 74.8% for *D. sargus*, being even higher than the average percentage of this study (15%). In contrast, pots seemed to be able to avoid *D. annularis* juveniles (average percentage of around 7%). Unlike studies in other areas (*Munro, 1974*; *Furevik & Løkkeborg, 1994*; *Hedgärde et al., 2016*), which concluded that larger pots are more effective than small ones, in this study, pot size seemed not to affect catch efficiency. However, the number of pots that actually produced a catch, ranged between 38.5 ± 3.3%

for SPs to 42.5 ± 4.3% for LPs, stressing the need for increasing catch efficiency by using attractive baits.

The cost of a collapsible pot as the one tested in this study ranges from €50 (SP) to €100 (LP), whereas 100 m of the traditional trammel net used for the sea trials is around €200. Assumed that 100 m of nets would correspond to more than six pots (as stated in the Materials and Methods section), the alternative gear is more expensive. Nevertheless, whereas the set nets usually last a single season and are then too damaged to be repaired, these kind of pots last up to 2 years. Another advantage of pots is that they afford a more limited access to the catch, making them less subject to depredation by large predators than set nets (e.g., seal-and dolphin-safe fishing gear; *Königson, 2011*; *Königson et al., 2015*). Moreover, they provide a greater catch quality, because they generally do not damage the specimens caught (*Suuronen et al., 2012*; *Olsen, 2014*). In addition, even if trammel nets require a shorter fishing time than pots, fishers could set different pots gangs in order to alternate the retrieve, and thus to haul them daily.

With reference to discards, the greater amount caught by the trammel nets clearly produces a greater impact on the benthic community, since discard mortality is high (*Suuronen et al., 2012*). Moreover, the cleaning of the trammel net implies an additional time and labor on deck for fishers, since discards must be released or untangled manually (*Sartor et al., 2018*; *Szynaka et al., 2018*). In contrast, the removal of discards from pots can be done without significantly reducing fishing time and leaving high probability for the unwanted organisms to survive (*Suuronen et al., 2012*). Discarding is a major issue for fisheries management worldwide (*Tsagarakis, Palialexis & Vassilopoulou, 2014*). In the Mediterranean, the Common Fisheries Policy (CFP *Regulation (EU), 2013*) has introduced the obligation to land ("discard ban") all the individuals of the species with minimum legal size (MCRS, for example, those species reported in the Annex III of the Council Regulation (EC) No. 1967/2006), thus emphasizing the need to reduce discards (*Damalas, 2015*). The landing obligation is a matter of concern among fishers, which are facing difficulties related to storing and bringing to land the former discard, due to limited hold space, and to sorting time or personnel increasing (*Maynou et al., 2018*). The introduction of a new and alternative technology in a fishery, such as innovative pots, could help to achieve this goal of discard reduction only if it is acceptable to both fishers and fishery policies. In this context, fish collapsible pots are revealed to be: practical (i.e., involving no major changes to common fishers' practices), cost-effective (i.e., easy to use and not expensive to maintain, no waste of time for cleaning the gear), efficient (i.e., large spectrum of species caught) and enforceable (i.e., easy to control by inspection authorities).

With reference to bycatch of sensitive and protected species, during our study the trammel nets caught five specimens of the pelagic stingrays (*Pteroplatytrigon violacea*), a frequent event in several Adriatic fisheries (*Bonanomi et al., 2018*). This Elasmobranch species, usually discarded due to its scarce commercial value, is considered as a "Least Concern" species in the IUCN red list (*Baum et al., 2016*). Another bycatch species caught by the trammel nets was the twait shad (*A. fallax*), listed in Annexes II and V of the Habitats Directive (*EU Directive, 1992*) as requiring close protection. In addition, the

passive nets deployed in the central and northern Adriatic are responsible for the bycatch—maybe as many as thousands individuals a year (*Lucchetti, Vasapollo & Virgili, 2017b*)—of loggerhead sea turtles (*Caretta caretta*), which are listed in Annex IV of the Habitats Directive. In contrast, the pots did not capture any of these species, substantially due of the small size of the pot entrance compared with the larger size of animals such as stingrays and sea turtles. As regards *A. fallax*, its absence in the pots catch may depend on their position close to the bottom, whereas trammel nets can also intercept pelagic fish species (*A. fallax*, *Sardina pilchardus*, *Engraulis encrasicolus*, *Scomber japonicus* etc.) which for different reasons can be discarded (*Goncalves et al., 2007*).

## CONCLUSIONS

The innovative pots tested in this study seem to provide a sound alternative to the traditional trammel nets used in the Adriatic Sea, at least in spring and summer, as concerns the small-scale fishery targeting common cuttlefish. These pots do not require a different vessel rigging nor changes to the on board practices; moreover, they can be used without baits and their foldable design involves that they can be easily stored on board the typical artisanal boats used in Mediterranean SSFs. The findings of this pilot study, although not conclusive, clearly indicate that these alternative gears go some way towards reducing bycatch and discards in SSFs while maintaining the commercial catch. Similar tests should be extended to other areas and seasons, also using baits, to provide a clearer assessment of their efficiency.

## ACKNOWLEDGEMENTS

This study does not necessarily reflect the European Commission's views and in no way anticipates future policy. This work was finalized under the project "Innovative technologies and sustainable use of fisheries and biological resources in the Mediterranean sea" (FISHMED-PHD). We are indebted to the personnel of CNR-IRBIM (Ancona) and CESTHA (Marina di Ravenna), to the crew of FVs Jessica (Portonovo), Zio Lino (Senigallia) and Nemo (Marina di Ravenna) for their help in fieldwork. We are grateful to Word Designs for the language revision.

### Funding

This study was conducted with the contribution of the LIFE Financial Instrument of the European Community, Tartalife Project - Reduction of sea turtle mortality in professional fishing (LIFE12NAT/IT/000937). The funders had no role in study design, data collection and analysis, decision to publish, or preparation of the manuscript.

### Grant Disclosures

The following grant information was disclosed by the authors:
LIFE Financial Instrument of the European Community, Tartalife Project—Reduction of sea turtle mortality in professional fishing: LIFE12NAT/IT/000937.

## Competing Interests

The authors declare that they have no competing interests.

## Author Contributions

- Andrea Petetta analyzed the data, prepared figures and/or tables, authored or reviewed drafts of the paper, and approved the final draft.
- Claudio Vasapollo analyzed the data, prepared figures and/or tables, authored or reviewed drafts of the paper, and approved the final draft.
- Massimo Virgili conceived and designed the experiments, performed the experiments, authored or reviewed drafts of the paper, stakeholder engagement, and approved the final draft.
- Giada Bargione performed the experiments, authored or reviewed drafts of the paper, and approved the final draft.
- Alessandro Lucchetti conceived and designed the experiments, authored or reviewed drafts of the paper, and approved the final draft.

## Data Availability

The raw data are available as a Supplemental File.

## Supplemental Information

Supplemental information for this article can be found online at http://dx.doi.org/10.7717/peerj.9287#supplemental-information.

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
