# Peer review of "Pots vs trammel nets: a catch comparison study in a Mediterranean small-scale fishery"

_PeerJ, doi:10.7717/peerj.9287_

## Round 0.1 · original submission · Major Revisions

Two expert reviewers have evaluated your manuscript and their comments can be seen below. Both reviewers have important suggestions about the manuscript that need to be dealt with in a revised version of the manuscript. Please pay particular attention to the issue raised by reviewer two regarding the statistical comparison of data involving distinct time periods.

·

Basic reporting

The reporting style is good with clear headings and concise text. The use of tables for data and results avoids an unwieldy results section. There are a couple of niche terms that are used (pots, soak time) that may need to be altered throughout to reach a wider audience. Citations are relevant and sufficient, although clarification is needed one of the comments made in association with the discard ban.

Experimental design

The experimental design is clear, and the aims of the study are clearly highlighted. The requirement for the research is highlighted. There are a differences in sample size and number of hauls at each location, these whilst accounted for in the analysis by standardising the CPUE data, still need explaining for the reasons behind the difference. The description of the gear type used is excellent and can be easily replicated by other researchers. The anlaysis is suitable and in line with other research of a similar field.

Validity of the findings

The validity of the findings and conclusions the authors have made are suitable and also the limitations of the research are clearly defined. The raw data is provided and of sufficient detail to suit the analysis conducted.

Additional comments

I have enjoyed reviewing your research and the relevance of such with the current agenda to reduce discards under the landing’s obligation. At sea sampling designs associated with commercial fisheries can often be difficult to achieve and this manuscript adds to this field well.
A minor point is the use of the word ‘pots’ to describe your alternative method to trammel nets. I recognise pots as a term used for predominantly crustacean fisheries, developed from the more traditional term creel. In this experiment I suggest the term ‘trap’ is used as this is more widely recognised when targeting fish (and cephalopods) as opposed to crustaceans. Fish traps are widely used across diverse fisheries globally, using the term ‘pot’ may restrict the research from reaching a wider audience.
Below are specific comments to the manuscript:
Abstract
Highlight the problem with high discard rates – many discarded species have high discard mortality.
Line 24: Change “more sustainable gears” to “a more sustainable gear type”
You discuss gill nets and trammel nets in the abstract, although they work the same way they are different, it is more suitable to discuss one net type in the abstract to avoid confusion.

Introduction
There is a bit of confusion when describing the gillnets and trammel nets, as I understand it, although both gear types “gill” the fish, a gillnet is a single sheet whereas a trammel net tends to be three sheets – a small inner mesh panel and two larger mesh outer panels. The description in the introduction needs to be clearer in highlighting the differences.
Line 43: Change “for about” to approximately”
Lines 58 – 60: Is there legislation that sets mesh sizes in these fisheries which may account for high discard rates?
Line 82: “pots have a low energy footprint” this statement is not necessarily true. There has been recent literature highlighting that in pot fisheries the carbon footprint can actually be higher than in mobile fisheries. See:
Abernethy, K.E., Trebilcock, P., Kebede, B., Allison, E.H. and Dulvy, N.K. (2010). Fuelling the decline in UK fishing communities? ICES Journal of Marine Science. 67(5), pp.1076–1085.
Arnason, R. (2007). The economics of rising fuel costs and European fisheries. EuroChoices. 6(1), pp.22–29.
Lines 102 – 106: It took several reads to understand what you are trying to say – this section could do with clarifying.
Material and methods
124 – 126: Do you think the spatial proximity of the GTR’s and fish traps may have affected the results?
Line 134 – 135: Why is the net length at Senigalla a different length – add reason for this.
Line 146: Again highlight why the number of pots differed
Line 147: Define soak time for the nonexpert reader
Line 165: Why was a Kruskal Wallis test chosen – if the data required a non-parametric test or violated the assumptions of a parametric test, this needs stating.
Line 179: What was the protocol used for AIC model selection – lowest value? If so, this needs stating.
Line 180: Was there any correction for soak time?
Results
Line 192: Change “the nets” to “GTRs” – maintain consistency in reporting
Line 198: Change “but especially” to “were more abundant”
Discussion
Line 237: There are many species that have a defined MCRS that are not subjected to the discard ban – Homarus gammarus, Cancer pagurus etc. Needs rephrasing
Line 246: What pots used in the area? By other fishers? Explain
Line 256 add a comma after “study”
Line 257 add a comma after “catch”. What is the value of these numbers? Individuals caught? CPUE? How does this stress a need for bait – clarification needed.
Line 262: Is there high discard mortality to increase the impact on the benthic community?
Line 270: Remove “in” after the word several
Line 271 – 272: This sentence is confusing, reword for clarity
279: Change “bigger” for “larger”

Figure 1: Why is there more points for M. Ravenna than the other sites? Was most of the sampling done at this site and only done once at the others? A smaller scale will highlight the sampling stations better as you have a good inset highlighting the general location.
Figure 2: The description in the methodology section of the trammel net design is excellent and really clear – this figure does not reflect the description. Maybe add a bit more detail to the legend, i.e. what the abbreviations stand for.
Figure 5: It would be helpful if the MCRS for each species could be plotted on the LFD so the reader can see the CPUE of the discards/catch between pot types.
Figure 6: As with figure 5. Does the dotted and dashed line in the shaded areas represent the mean of the data? What does the vertical dotted line at 12 cm denote on the D. annularis plot represent? A label plotted on both plots highlighting on which side of the 0.5 line the data represents Pots and which side represents GTRs.

Reviewer 2 ·

Basic reporting

I have not offered a detailed response in this section given the issue highlighted in section 3.

Experimental design

I have not offered a detailed response in this section given the issue highlighted in section 3.

Validity of the findings

The authors approach to data transformation is of significant concern and in the reviewers opinion will require comprehensive revision and full reassessment. Unfortunately I believe the current approach has undermined the paper and the fundamental validity of the results reported.

The authors have presented data transformation and standardisation of CPUE for GTR to 12 hrs per 1000m (L158 – L16) and Pots to 24 hrs per 1000m (L161 – L163). Whilst I appreciate the basic narrative and apparent attempt to align with industry working practices, this transformation of data to incomparable temporal scales is statistically unsound and will have fundamentally influenced the proceeding complementary analysis, weighting a significant negative bias against GTR catches in the subsequent comparisons against fish traps. As this data forms the foundation of the manuscript, this issue will have influenced all proceeding data analysis, results and the conclusions presented.

I would strongly recommend the authors redevelop their analysis and compare catches from both gears to a standardised period prior to any resubmission. The scientific question and driver behind this study is about the relative catch efficiency of different gears types, not actual industry working practices, however this could be assessed through a seperate complementary analysis within a revised manuscript.

---

## Round 0.2 · Major Revisions

Two expert reviewers have evaluated your revised manuscript and they have some important observations which need to be attended to in a resubmission if you choose to do so.

Reviewer 3 ·

Basic reporting

Some literature references should be add in the Introduction section.

Experimental design

Standarization of captures should be carefully reviewed.

Validity of the findings

Some results should be added and clarified.

Additional comments

This study contributes to the general interest to the advancement of applied fisheries science. Discard amounts is currently a major concern for the EU Policy and the development of new techniques aimed at reducing discards is necessary.
However, several aspects need be improved in the manuscript. Mainly, Results section should be clarified and some additional results should be added.
Abstract:
Line 30: the two most abundant commercial species.
Introduction:
Line 46-49: Information about how set nets catch fish seems unnecessary.
Line 52: Replace “Amengueal-Ramis et al., 2016” by “Amengual-Ramis et al., 2016”.
Line 57: You could add other studies about discards of trammel nets in Mediterranean sea.
Line 62: Here you could also try to add other references.
Line 68-72: Also the Common Fishery Police in the European Union has focused on the discard reduction. You talk about in the Discussion section but you could add some information here too.
Line 110-112: You should add more details about the aims of the study. For example, to test the effectiveness of the pots reducing discards, …
Materials and Methods:
Line 122-123: The information about the bottoms that appears in the text is not the same as in the Table 1. What is the correct information? Also, you do not need to provide this information in both places (text and table), it is too repetitive.
Line 125: Three different vessels in the three study areas?
Line 126: Why “traditional trammel net” is abbreviated as “GTR”, not as “TTN”?
Table 1: In addition to the mean value (AVG), you should add a standard deviation, standard error or range.
Line 127: I’m not sure the term “set” is clear enough. Do you mean “fishing trips” or “fishing journeys”?
Line 135: Is the diameter of float line 5.5 mm as you say in the text or 8 mm as you say in the Figure 2?
Line 137-138: The sentence “the hanging ratio of...” is repetitive because you repeat all this information in the Figure 2. You could delete it.
Line 139: “and 300 m at Senigallia and”. Delete the second “and”.
Line 141: Is “Trapula” the name of the factory? Clarify this.
Line 154: Each fishing journey, what do you haul? A trammel net, a line of small pots and other line with large pots? Clarify.
Line 157: Change by “invertebrates and fish species with no commercial...”.
Line 163-170: I have some doubts about the standardization. I understand that different gears need different soaking times. However, if the pots are commonly retrieved after 2-3 days, why you standardize in 24 hours? And other reflexion, I wonder what is the reason for leaving pots during 2-3 days, if they obtain an amount of commercial captures similar than the trammel nets just in 24 hours? Does the catch really double in 48 hours? Maybe pots need more than 48 hours to start capturing target species, because the first species captured act as a claim. If that is the case, and captures do not occur linearly, standardization in 24 hours would not be valid. If a wide range of soaking time is available, you can test the trend (amount of captures vs soaking time).
Line 175-185: You could clarify this part. What type of pot? Do you analyze all the pots together? Also, you should define each parameter in the equation.
Supplementary Tables: In line 157, you say that discards are that species with no commercial value and individuals under the MCRS. However, in Supplementary Tables, individuals under the MCRS are included in the commercial catches. You should separate the portion that belongs to discarded fraction.
Also, you could highlight (maybe in bold) in the Supplementary tables, the most important species commented in the text.
Line 202: Change “important” by “abundant”.
Line 207-208: S. umbra at Marina di Ravenna is mentioned twice.
Line 210: You should indicate that the mean biomass value is calculated as CPUEw.
Table 2: In the Table caption, you should indicate that the values presented are the mean value, +/- standard deviation or standard error, and the confidential intervals in brackets.
Also, the total values (shown as a separate table) could be added under the three study sites, as a new file.
Line 211: “did not overlap” is not correct. The CI values of the three gears overlapped.
Line 214: The same analysis made with the CPUEw of the total commercial catch, could be made with the CPUEw of the most important species (almost with cuttlefish).
Line 221: Again, specify CPUEw in the mean biomass values.
Line 228: “The lines of LPs and SPs were mostly above those of GTRs, indicating a greater catch efficiency.” That is not clear in Figure 5. You should apply an statistical analysis with the CPUEw or CPUEi.
Figure 5: In figure caption, specify the name of the sites. Also, in the legend, replace BP with LP.
Line 231: “were equally efficient only for the larger sizes”. I’m not able to see that in the Figure 6. For me these results are a bit confusing. You could explain more these results. You could also add other more simple analysis, for example, an one way analysis of the length between the three gears. You could also indicate the percentage of undersized D. annularis captured by each gear.
Line 234: “GTR catch curves”? Rewrite this sentence. I wonder if you could add a curve for GTR catches or at least represent its CI.
Discussion:
Line 242: “(CFP Regulation (EU), 2013/1380)(EU Regulation 1380, 2013)”. Correct this.
Line 243: Delete one “of”.
Line 245: You could add some information about the consequences of the landing obligations for the fishers. For example, see Maynou et al 2018, Fishers’ perceptions of the European Union discards ban: perspective from south European fisheries.
Line 253: What does “new pots” means? Are pots similar as used in this study? Also, link this sentence to the previous paragraph.
Line 256: “A significant difference between trammel nets and pots was found for annular seabream”. This is not shown in the Results section.

Reviewer 4 ·

Basic reporting

The paper of Petetta et al. concerns an interesting work aimed at testing the catch efficiency of two type of pots (still not used in Mediterranean) to reduce discards, to be used as an alternative of set nets.
The work is of interest, because it deals with a current topic, e.g. the reducion of discards (see the current management policies on discard ban) and, more in general, with the reduction of the fishing gear impacts and the related socio-economic implications.

This manuscript is a resubmission, following the revision of two referees.
After a careful revision (also of the rebuttal letter), in my opinion the new version submitted by the Authors fully meet the comments made and the explanations are generally satisfactory.

The applied methodology (see below) is correct, the presentation and discussion of the results is clear and unambiguous.

I made a detailed revsion of the text, tables and figures (including the supplementary tables), including several corrections and comments.
I suggest to carefully revise the text accordingly.

Experimental design

The Authors reported that the text was aimed at replacing as much as possible commercial conditions: I agree on this approach but I suggest to explain clearly (at the beginning of Material and Methods) this aspect: e.g. reporting that the characteristics of the gears (length of GRT and GNS), fishing grounds, fisging operations (fishing time), sorTing of the catch, were carried out following the fishermen procedures, without interferences from the scientists on board.
Of course this approach implied the calculation of different CPUE for the different gears used (I agree that it is not possible to use the same CPUE for gears operating in different ways): I suggest to mention this aspect in the discussion of the results.

Validity of the findings

In my opinion the findings of this study are interesting and can represent an important information for he management of the small scale fisheries and of the coastal fishery resources.

Additional comments

I attach a revised version with corrections and comments of
- the text (I used the Word revised version - I used the track change)
- the tables and the figures (in the pdf file of the manuscript)
- the two supplemetnary tables (with track changes)

Annotated reviews are not available for download in order to protect the identity of reviewers who chose to remain anonymous.

---

## Round 0.3 · Minor Revisions

One reviewer has re-evaluated your manuscript and has some suggestions for its improvement.

Reviewer 3 ·

Basic reporting

Some minor modifications are nedeed.

Experimental design

No comment.

Validity of the findings

Some clarifications are nedeed.

Additional comments

The paper has been improved. However, there are still few clarifications needed.
Line 25: Replace “SFFs” with “SSFs”.
Line 37: Sometimes, throughout the manuscript, the term “bycatch” is used (also in Discussion Line 295); however in the Results section, the terms used are “landed species” and “discarded species”. You should clarify this because “bycatch” and “discards” are not synonymous.
Line 72: Please, remove the excess “the”.
Line 104-113: Review the aims of the study. “iii) to widen the range of commercial species to be caught in comparison to the traditional pots used in the area;”. A proper comparison of the captured species by each type of pot is not performed in this study.
I propose something similar to “Based on these considerations, a pilot study was devised to test the effectiveness of a fully collapsible pot design in comparison of a traditional set net under commercial conditions; in terms of i) catching commercial species, iii) reducing discards, and iii) other aspects related to use and handling.”
Line 112: Replace with “these areas”.
Line 117-118: You say previously that the study is conducted in three areas. So, you could add the name of the three areas there: “and involved three coastal areas (Marina di Ravenna, Portonovo and Senigallia), where depth ranges from 5 to 19 m (Figure 1).”
Also, you should move the sentence “Specifications of bottom type and average depth of the three sites are listed in Table 1.” behind the sentence “and involved three coastal areas (Marina di Ravenna, Portonovo and Senigallia), where depth ranges from 5 to 19 m (Figure 1).”
Table 1: The SD in depth should be added.
Line 146: Please, move the reference at the end of the sentence.
Supplementary Table 1 y 2: These tables have greatly improved their clarity. Only in the column headings you use “Big CPUEW” and “Small CPUEW” instead of “LP CPUEw” and “SP CPUEw”.
Also, I insist again on a point already discussed in the previous review. You say (line 157-160) “The total catch for each gear was thus divided into a landed catch (species with commercial value, not necessarily target species) and discards, i.e. species discarded for different reasons (invertebrates and fish species with no commercial value, commercial individuals under the MCRS or in poor conditions)”. So, why do you include the commercial individuals under the MCRS as landed or commercial (i.e., marketable) catch? They should be included as discarded species in Table 2. And, are this fraction (i.e., commercial individuals under the MCRS) included as landed catch in all analyzes carried out on the manuscript? If so, the results obtained are not entirely correct. I understand that, at this point, to modify this is a great effort, but I encourage the authors to carry out this modification since it would give more reliability and robustness to the results.
Line 208-210: “Other landed species caught by the pots were S. umbra, Conger conger (LP) and S. umbra (SP) at Marina di Ravenna and Dentex dentex (SP) at Portonovo (Supplementary Table 1)”. Please, review this sentence. S. umbra at Marina di Ravenna is named twice.
Line 230-231: Pots will be a valid alternative to GTRs, just if pots are equally or more economically profitable. The profit of the fishery depends mainly on the catches of S. officinalis (the most abundant capture and with highest price (I suppose)). Therefore, this is a very important point: “The catch comparison curves (Figure 6 and Table 3) demonstrate that for S. officinalis, pots (both dimensions) were more efficient than GTRs”. However, it is not clear enough if this higher efficiency is limited only to certain sizes. Could you clarify this? Maybe an additional analysis could be needed with the CPUEw of cuttlefish.
Line 252-259: The main results obtained in this study should be added in this paragraph. Also, you could discuss about the economic efficiency of the pots vs GTRs.
Line 262: Replace “weigth” with “weight”.
Line 260-263: This part is referred to discarded fraction. So, you should move to the paragraph that begins in Line 275; specifically, when you talk about the obligation to land. Also, you could add the percentages obtained in this study.
Line 295: Review the use of the term “bycatch”.

---

## Round 0.4 · accepted · Accept

I am satisfied with the changes made to the manuscript.